# Host’s Immunity and *Candida* Species Associated with Denture Stomatitis: A Narrative Review

**DOI:** 10.3390/microorganisms10071437

**Published:** 2022-07-16

**Authors:** Pierre Le Bars, Alain Ayepa Kouadio, Octave Nadile Bandiaky, Laurent Le Guéhennec, Marie-France de La Cochetière

**Affiliations:** 1Department of Prosthetic Dentistry, Faculty of Dentistry, Nantes University, 1 Place Alexis Ricordeau, 44042 Nantes, France; ayepa_alain@yahoo.fr (A.A.K.); octave.bandiaky@univ-nantes.fr (O.N.B.); laurent.leguehennec@univ-nantes.fr (L.L.G.); 2Department of Prosthetic Dentistry, Faculty of Dentistry, CHU, Abidjan P.O. Box 612, Côte d’Ivoire; 3EA 3826 Thérapeutiques Cliniques Et expérimentales des Infections, Faculté de Médecine, CHU Hôtel-Dieu, Université de Nantes, 1, rue G. Veil, 44000 Nantes, France; mfcochet@hotmail.com

**Keywords:** *Candida* spp., dental plaque biofilm, denture-related *Candida* stomatitis, inflammation, mucosal immunity, aging, receptors, Ig A

## Abstract

Denture-related *Candida* stomatitis, which has been described clinically in the literature, is either localized or generalized inflammation of the oral mucosa in connection with a removable prosthesis. During this inflammatory process, the mycobacterial biofilm and the host’s immune response play an essential role. Among microorganisms of this mixed biofilm, the *Candida* species proliferates easily and changes from a commensal to an opportunistic pathogen. In this situation, the relationship between the *Candida* spp. and the host is influenced by the presence of the denture and conditioned both by the immune response and the oral microbiota. Specifically, this fungus is able to hijack the innate immune system of its host to cause infection. Additionally, older edentulous wearers of dentures may experience an imbalanced and decreased oral microbiome diversity. Under these conditions, the immune deficiency of these aging patients often promotes the spread of commensals and pathogens. The present narrative review aimed to analyze the innate and adaptive immune responses of patients with denture stomatitis and more particularly the involvement of *Candida albicans* sp. associated with this pathology.

## 1. Introduction

Between 15% and 70% of denture wearers have *Candida* stomatitis related to their removable prosthesis [1]. The prevalence of this pathology is preponderant among hospitalized elderly people [2], smokers [3], and people with associated affections such as diabetes [4]. This disease also called “denture-related *Candida* stomatitis” (DRCS), and is considered an infectious inflammation, different from sterile inflammation, characterized by mechanical denture stress [5] and an imbalance of the oral microbial flora or dysbiosis [6]. Denture wearers have a less diverse oral microbiome than dentate patients. This is simultaneously reflected in the abundance of opportunistic commensals such as *C. albicans* and the proliferation of bacterial species revealed by next-generation sequencing (NGS) [7,8]. Moreover, *C. albicans* can penetrate the resin, thereby constituting a microbial reservoir [9,10].

Under these conditions, the presence of a removable prosthesis within the oral cavity, particularly in immunocompromised older individuals, causes environmental modifications, favoring the colonization and transformation of the pathobiont *C. albicans* from the yeast stage to the hyphal form characteristic of fungal pathogenesis [11]. This reversible transformation of *Candida* morphology is favored by several environmental parameters characteristic of the buccal cavity, such as temperature (>36 °C), pH acidity, access to nutrients (iron, glucose), the presence of serum, and high levels of CO_2_ [12].

Furthermore, in the presence of infection, if *C. albicans* passes into the bloodstream, several organs such as the kidneys, liver, and spleen may be affected [13]. In addition, *C. albicans* infection may contribute to the process of carcinogenesis, so a possible correlation between *Candida* infection and potentially precancerous oral diseases such as oral squamous cell carcinoma has been suggested by several authors [14,15,16].

The clinical diagnosis is based on the classification of DRCS, also called “histologically chronic erythematous candidiasis”, which is categorized into three types: Type I is simple localized inflammation; type II is diffuse edematous erythema; and type III is hyperplastic papillary granulomatous inflammation of the palatal mucosa [17].

Other associated clinical forms such as angular cheilitis are related to DRCS [18].

Thus, in the presence of denture stomatitis, bacterial and fungal diversity is altered. Dysbiosis of the oral microbial plaque can lead to a breakdown in the homeostasis of the complex and dynamic relationship with the host’s immune system [19]. In this mini review, we analyzed and discussed the host immune response to *Candida* infection in the oral cavity.

## 2. DRCS and Bacterial—Fungal Dysbiosis of Oral Microbiota

Poor maintenance of the prosthesis causes accumulation of biofilm, colonizing the denture surface with more than 10^11^ microorganisms (bacteria, fungi, viruses) per gram of dry weight [20,21]. During this process, the immune responses of individuals are particularly refined to control the bacterial and fungal populations of *C. albicans* and non-*albicans* (*Candida glabrata*, *Candida tropicalis*, *Candida krusei*, *Candida parapsilosis*, *Candida dubliniensis*) [22]. The diversity of oral microbiota decreases and for three quarters of denture wearers, *C. albicans* easily proliferates and changes from reversible plasticity (harmless unicellular form) to pseudohyphae and hyphae that induce mucosal bacterial dysbiosis [23,24]. In particular, *C. albicans* colonizes the surface of the mucosa, but preferentially the surface of the prosthetic base [25]. This is due to the fact that the epithelial cells can discriminate between commensalism and pathology in the presence of *C. albicans*.

Additionally, the microbiome in the form of a biofilm promotes cross-kingdom interactions between bacteria and fungi that influence the growth, morphogenesis, and drug resistance of *C. albicans* [26]. The prosthesis in the oral cavity creates ecological niches in which sessile cells of *C. albicans* have the propensity to form biofilms thanks to their plasticity by adapting to the underlying regulatory network [27]. *Candida* selects its bacterial cohabitation; thus, the acidity of its environment is favorable to bacilli (streptococci and lactobacilli) but unfavorable to the fusobacteria, bacteroidia, and flavobacteria classes that do not tolerate acidity well [28]. Thus, the theory that several microbes besides *Candida* participate in DRCS holds true. Another hypothesis rests on the fact that in the absence of DRCS, a sufficient immune response maintains the health of the palatal mucosa, despite a high candidal presence on the surface of the epithelium.

The in vitro confirmation of this relationship is demonstrated by the culture of *Candida* spp. alone or in association with bacteria on the surface of the resin prostheses. *Candida* cultivated alone does not develop extensively in the form of hyphae, while in association with *Streptococcus sanguinis*, *Streptococcus gordonii*, *Actinomyces odontolyticus*, and *Actinomyces viscosus*, *Candida* significantly proliferates in a hyphal form. On the other hand, if *Porphyromonas gingivalis* is added to this group of bacteria, there is a decrease in the virulence of candidiasis leading to a decrease in hyphal production [29,30]. Locally, within the biofilm the development of hyphae is promoted by the interaction with components of the bacterial cell wall such as peptidoglycan [31]. Another example of a positive cooperation between *C. albicans* and a bacterium such as *Streptococcus oralis* was presented by Xu et al. [32]. This association facilitates the degradation of E-cadherin at the level of the epithelial junctions and enables the penetration of microorganisms in the mucosa. Similarly, coaggregation and a mutualistic relationship between *C. albicans* and *Staphylococcus aureus* promote adhesion to both the mucosal and denture surfaces [33]. This synergistic cooperation between the two microorganisms leads to an increase in their pathogenic potential [34]. Thus, in co-culture, *C. albicans* and *S. aureus* produce hydrolytic enzymes (SAP: secreted aspartyl proteinase) that contribute to proteolytic activity [35].

Interestingly, lactate production by another co-commensal in the oral cavity, *Streptococcus mutans*, provides carboxylic acid substrates that are sufficient to promote *C. albicans*-mediated alkalinization of the microenvironment [36]. In turn, *C. albicans*, by producing a high amount of farnesol, promotes the colonization of *S. mutans* [37].

Regarding intra-kingdom interactions, there is no consensus on the relationships between *C. albicans* and *C. glabrata*. Some authors lean toward a synergy between the two fungi [38], while others have not found evidence of any cooperation that could favor DRCS [39]. *C. dubliniensis*, which is found in 10% of DRCS cases, is another fungus that is closely related to *C. albicans* [40]. This can be explained by the fact that most of these results do not accurately reflect the complexity of the composition and interactions in vivo within the oral microbiome.

Another aspect of DRCS concerns the analysis of the proportion of salivary load between *Candida* and bacteria in people with removable prostheses. In the study by Kraneveld et al. [41], the authors found that among 82 patients (aged 60–80 years) wearing a partially or completely removable prosthesis, 97% tested positive for the internal transcribed spacer (STI) characteristic of the fungus. These authors demonstrated a decrease in the diversity of the salivary microbiome with a difference between totally edentulous and dentate individuals. However, this difference was not correlated with the presence of *Candida* [41]. On the contrary, another study, based on the culture of *Candida* spp., showed in a population of 123 patients fitted with orthopaedic devices a high candidal population that can lead to an imbalance of the oral microbiome [23]. Confirmation was provided by Fujinami et al. in a sample of 18 dentures wearers (mean age, 80.3 years). Based on measurements of *C. albicans* DNA concentrations and bacteria by real time PCR, this study showed the abundance of the genera *Streptococcus*, *Lactobacillus*, *Rothia*, and *Corynebacterium* on the surface of removable dentures compared to dental plaque. *C. albicans* was positively correlated with these acidogenic bacteria [42]. Nevertheless, the commensal *C. albicans* can become a virulent pathogen under certain conditions.

## 3. Denture Plaque and *C. albicans* Virulence

During the evolution of cohabitation with humans, the commensal *C. albicans* has acquired mechanisms that have given it the ability to hijack the innate immune system [43]. It can adapt to different environmental conditions of its host, express virulence factors, and thus cause infection. Among these conditions, the presence of a removable prosthesis, alone or in combination with other local (lack of hygiene, sugar consumption), general (immunodeficiency), and medicinal (antibiotic intake) parameters favors the colonization of *Candida* spp. [44] (Table 1). In the absence of hygiene, the consumption of sugar leads to a drop in pH causing selective pressure on the microbiome and a decrease in its biodiversity. This facilitates the growth of *Candida* spp. [45], which can thus integrate into different oral niches [46].

The transformation of *C. albicans* involves different signaling circuits such as glucose accessibility (conditioned by processes such as the cyclic AMP/protein kinase A (cAMP/PKA) pathway [52] and transcriptional regulation of biofilm formation including *C. albicans* on an abiotic surface. The latter process depends on several factors such as enhanced filamentous growth protein 1 (Efg1), which is a *C. albicans* gene involved in the filamentous form [53,54]. Other factors, such as Cph1 (Cyanobacterial Phyto-chrome 1) and Czf1 (zinc finger cluster transcription factor), promote hyphal development, while hyphal gene expression is inhibited by transcriptional repressors, including the Hog1 (high osmolarity glycerol) and Rfg1 (repressor of filamentous growth 1) pathways [55,56].

However, both morphologies, yeast and hyphae, are inseparable from the virulence phenomenon of *C. albicans* [57]. Thus, cells in the form of yeasts are readily disseminated, and hyphal cells are readily penetrated [58]. The potential virulence factors of *Candida* in the presence of a removable prosthesis are also related to various mechanisms such as adhesion, invasion, and production of exotoxins, enzymes, interactions with metabolism, or factors counteracting the host’s immune defense systems (Figure 1).

### 3.1. Adhesion of C. Albicans to Prosthetic Abiotic and Epithelial Biotic Surfaces

*C. albicans* can adhere to epithelial surfaces but also to the prosthetic base as well as to other eukaryotic and prokaryotic microorganisms of the florid biofilms [59]. Thus, the cohabitation of eukaryotic and prokaryotic pathogens makes DRCS difficult to treat, as it requires complex multidrug treatment strategies.

In the presence of these different surfaces, *Candida* develops hyphae that produce three family of adhesins [60,61].

The first family of adhesins is the agglutinin-like sequence (Als) genes, the second family is the hyphal wall proteins (Hwp), and the third is a member of the HYR gene family that are secreted by hyphae [60,61,62]. By contrast, yeast wall protein 1 (Ywp1) appears to promote candidiasis dissemination by opposing the phenomenon of adhesion [63]. On acrylic prosthetic surfaces, the adhesion of *Candida* spp. depends on van der Waals and electrostatic forces. A further hydrophobic attraction between *Candida* and inert surfaces can occur under short distances, which favors adhesins processus [64]. This latter parameter is considered to be an ancient damage-associated molecular pattern (DAMP) that initiates innate immune responses [65].

### 3.2. Candida Invasion Mechanisms

The mechanisms of fungal invasion into epithelial cells start by adhesion of *Candida* cell wall proteins Als3 and Sas1 to E- and N-cadherins of epithelial cells, followed by the invasion pocket of hyphae, which grows and provides virulence factors [66,67].

It is in this invasive form of hyphae that *C. albicans* activates the three mitogen-activated protein kinase (MAPK) pathways (namely, p38, JNK, ERK1/2), the nuclear factor kappa-light-chain-enhancer of activated B cells (NF-B) pathway, and the phosphatidylinositide 3-kinase (PI3K) pathway that trigger the immune reaction [68,69,70,71].

### 3.3. Candida Exotoxin Release and Host Damage

*Candida albicans*—through the secretion of several hydrolytic enzymes including secreted aspartyl protease (SAP), phospholipase, and hemolysin—can attack and degrade host membranes of mucosal surface cells [72,73].

Furthermore, the extent of cell elongation protein 1 (Ece1p) of *C. albicans* participates in the pathology of the oral mucosa by activating innate immunity in vivo. This also expresses candidalysin, a pore-forming α-helical peptidetoxin that is encoded by the ECE1 gene [67,74,75].

### 3.4. The Immune Evasion of C. Albicans

Naturally, to limit the proliferation of microorganisms on the surface of the mucosa and their penetration into the tissues, the host has several means of protection: the epithelial barrier and its innate and adaptive immune system.

Recognition of *C. albicans* as commensal precedes the immune response [76]. The patterns of DRCS are recognized by epithelial cells and cells of the immune system residing in the tissues (mast cells, macrophages, and dendritic cells) expressing pattern recognition receptors (PRRs) [77]. These PRRs are the link between the host’s immune responses and protection against pathogens [78].

However, PRR ligands are produced both by the resident microbiota in healthy patients and by pathogens. These PRRs, the paradigms of immunology, are the link between microbial symbionts and their hosts. Before the clinical onset of pathology, PRRs are able to detect microbial molecules in order to initiate inflammatory responses. From a certain threshold, the host triggers an innate immune response to *C. albicans*, and PRRs maintain communication with the microorganism commensals while participating in a beneficial cohabitation within the microbiota. However, through PRRs (e.g., Dectin 1 and TLRs), cells within the oral mucosa can detect pathogen-associated molecular models (PAMPs) that recognize molecular structures expressed by invading pathogens, for example, β-glucans or mannans such as the components of the cell wall of *C. albicans* in the form of yeast and hyphae [13,79], (Figure 2) and such as lipopolysaccharides (LPS), fimbriae, and bacterial flagellin [80]. Following the interaction between PRRS and PAMPS, the release of a panel of cytokines/chemokines and specific proteins will activate adaptive immunity through T lymphocytes. In turn, these T cells travel to the infected site where they secrete cytokines to stimulate candidal phagocytosis of macrophages and neutrophils.

In the same way, *C. albicans* in the denture environment can bypass the host’s defenses and even resist antifungal treatments [81]. A dynamic balance persists between the host and the harmless unicellular fungus in healthy individuals. However, the lack of effective immune surveillance facilitates fungal growth in the form of hyphae [82,83,84,85]. The denture promotes an overgrowth of *C. albicans* and numerous commensal bacterial species. *C. albicans* in the form of hyphae crosses the epithelium of the mucosa and enables bacterial penetration (Figure 3).

## 4. Innate Immune Responses and Denture-Related Candida Stomatitis

The epithelial cells and immune cells residing in the tissues play the role of “alert” cells, since they react quickly following the detection of danger [86]. This danger triggers the activation of the cells, which then secrete inflammatory mediators (histamine, pro-inflammatory cytokines, chemokines, lipid derivatives), themselves responsible for activating endothelial cells and the initiation of the vascular phase of the inflammatory response [87] (Figure 4).

Cytokines: Tumor necrosis factor alpha (TNFα), Interleukines (IL-1, IL-6, IL -8, IL-17, IL-22, IL-23, IL-36), granulocyte colony stimulating factor (G-CSF), gamma interferon (IFN-Υ). Chemokines: RANTES, IL-8, MIP3a; ROS, reactive oxygene species; RNS, reactive nitrogen species; NET, Neutrophil Extracellular Trap; AMPS, Antimicrobial Peptides substances (β-defensins, cathelicidin (LL-37).

Innate recognition

Early recognition: Under the denture, following the appearance of hyphae and tissue penetration of *C. albicans*, the monocytes of oral mucosa (neutrophils, macrophage, and dendritic cells) appear in order to react and govern the T cell antigens towards fungal aggression. Mucosal keratinocytes and myeloid cells can identify commensals and pathogens, via expressing families of pattern recognition receptors (PRRs) classified according to protein domain homology or the cellular localization:
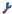
 1.TLRs, Toll-like receptors.
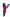
 2.c -type lectin receptors, CLRs (Dectin 1 and Dectin 2 recognize β-glucans in the cell wall of *Candida*); Mincle, macrophage-inducible C-type lectin; MR, recognizes mannose receptor; DC-Sign.
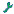
 3.EphA2 (Epithelial Ephrin type-A receptor 2) recognizes β-glucan [88,89].

Epithelial cells activate mitogen-activated protein kinase 1 (MAPK1)- and c-FOS-related pathways to mediate cytokine production.

DAMP (danger-associated molecular patterns): hydrophobicity and alarmins. Under denture DCRS, sustains necrotic or apoptotic cell death due to the release of endogenous ligands such as DAMP [65].

Adaptive antifungal responses

Naive peripheral T cells are a mature subpopulation of regulatory T cells present in TC. They keep the commensal balance between inflammation and adaptive response. The mature dendritic cell with: 
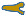
 MHC-II. MHC class II molecules present antigenic fragments acquired in the endocytic route to the immune system for recognition and activation of CD4+ T cells.

The Naïve T cell recognizes these antigens through their Toll cell receptors (TCRs) 
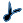
 and activates the Th1, Th2, Th17, and memory cell. Th17 cells secrete the IL-17 and IL-22 family of cytokines. The IL-17 act in return on the epithelial cells and regulate the expression of genes in relation to AMPS. The IL-22 act on the neutrophils [90,91].

From Th1, interferon (Inf-Υ) can activate the macrophage which has engulfed *C. albicans*, and cause its cell death. It is thanks to the inflammasome 
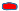
 (nucleotide-binding domain and leucine-rich repeat receptors (NLRP3) that the fungus is able to trigger pyroptosis [92,93].

### 4.1. Cytokine Mediators and Denture-Related Candida Stomatitis

It is well established that epithelial cells produce a variety of cytokines in response to the presence of *C. albicans*, including granulocyte-macrophage colony-stimulating factors (GM-CSF, G-CSF), IL-1a, IL-1b, and IL-6, along with the chemokines RANTES, IL-8, and MIP3a [68,94,95]. It seems that both the capacity and threshold of colonization by *Candida* are preponderant. At the salivary level, for prevention, the quantities of cytokine interleukins (IL-6, IL-4), c–c chemokine ligand 3 (CCL-3), and transforming growth factor-beta (TGF-β) vary upward in aging denture wearers with DRCS [77]. On the other hand, salivary IL-12 decreases in the same patients [89]. In the same group of aging denture wearers, it was found that serum cytokines (IL-6, TNF-α, IL-4, IL-10) increased and could prevent the complications of stomatitis [96,97,98].

Furthermore, in another in vitro study, IL-18 gene expression was up-regulated during mixed-species (bacteria and fungi) infection [99]. This gene corresponding to the pro-inflammatory cytokine IL-18 influences the activity of monocytes and promotes the phagocytosis of microbial antigens. In turn, these same stimulated monocytes secrete specific cytokines such as IL-23 whose quantity increases with the presence of *C. albicans* and bacterial LPS [100,101].

A comparative study of the peripheral blood from 20 patients with DRCS and 24 control patients made it possible to highlight lymphocyte profiles by flow cytometry and to evaluate the production of cytokines by T lymphocytes. No significant difference was found between the two groups. These authors evoke the hypothesis of the limited capacity of patients with dentures and older patients to fight against the infection [102]. Generally, the process of aging clearly influences several factors in the blood (serum IL-4 and interferon-gamma (IFN-Υ)), independently of the presence or absence of DRCS [77]. Other authors [103] using enzyme-linked immunosorbent assay (ELISA) did not systematically find an association between salivary pro-cytokines (IL-6, IL-8, IL-10, IL-17), intercellular adhesion molecules ((ICAM-1), TNF-α), DRCS, and patient age.

By contrast, a study of the salivary IL-6 GG genotypes in different clinical classes of DRCS found a significantly increased expression in Newton classes I and II (*p* ≤ 0.01) compared with class III. For these authors, the significant differences in some genotypes of TNF-α, TGF-β, and IL-6 in DRCS patients can contribute to our understanding of the host defense [104]. Another point of view expressed by many authors is that DRCS is more dependent on predisposing factors such as the nightly use of removable prostheses and/or poor oral hygiene rather than merely on the presence of *C. albicans* [1,105]. It seems that the diversity of cytokine receptors depends on the individual response and conditions including the quantity of cytokines. In the same vein, an interesting study analyzed the role of Th1-/Th2-type salivary cytokines in the saliva of HIV patients with DRCS. This study focused on seronegative (HIV) removable denture wearers with and without DRCS and found no significant differences in the level of Th1/Th2 cytokines between the two groups. Thus, in this case, the Th salivary cytokines of these seronegative patients do not directly influence DRCS [106].

More recently, flow cytometry analysis of the cytokines (IL-2, IL-4, IL-6, IL-10, IL-17a (IFNγ; TNF-α)) of 93 patients with removable dentures, including 42 with DRCS, showed a significant upsurge in Th1 (IFN-γ, IL-2), Th2 (IL-4), Th17 (IL-17a, IL-22), and Tregs generated (IL-10) in the the latter group (Figure 4). *Candida* culture was positive in 48 of the 93 individuals including 29 with DRCS. This latter group had a significantly higher number of isolated *Candida* on the intrados of the prosthesis (*p* = 0.0113). However, a significantly (*p* = 0.03) elevated quantity of *Candida* on the palatal mucosa of DRCS-free patients was also detected. It therefore seems that an appropriate immune response is triggered in patients with DRCS, but even in the absence of DRCS, the prosthesis maintains a candidal reservoir. Thus, in the presence of DRCS, the inflammatory response of the palatine mucosa makes it possible to combat the increased presence of *Candida*. However, this response is ineffective against *Candida* colonization on the prosthesis [101].

This fact was confirmed through a study with genetic mouse models, which showed early inflammation of fungal origin outside the influence of immunosuppression. Thus, in this situation, the host’s type-17 protective immunity remains. This suggests that persistence of *C. albicans* in oral mucosal tissues does not directly depend on antifungal immunity [107]. Additionally, diversity in *C. albicans* intraspecies can trigger specific, time-limited responses, allowing them to transition from commensalism to pathogenicity independently of the host [76].

### 4.2. Complement System and Denture-Related Candida Stomatitis

In the presence of inflammation (DRCS), lysis via complement is one of the host cell recognition systems that involves the complement receptor CR2/CR3 of *C. albicans* [43]. The opsonization of *C. albicans* involves the C3 complement and C5 activates phagocytosis by response to pro-inflammatory cytokines of fungi [108,109,110,111]. In both class II and class III DRCS, C3 is not regulated. C3 in combination with integrin α-M/β2 participates in the adhesion of macrophages and monocytes. However, *C. albicans* has the ability to bypass the action of C3 by mimicry of the C3 receptor [112]. On the other hand, this opsonization can be inhibited by non-specific binding of glucose to lysine residues (glycation) at the active site of complement C [112,113]. This partly explains why unbalanced diabetes is a risk factor for DRCS. The fungal wall, due to its polysaccharide constitution, is a powerful activator of complement via the alternate route (Figure 4) but *C. albicans* may reduce the involvement of the complement system in inflammation. This is because *C. albicans*, by producing proteins on its surface, can decrease the efficiency of the complement system [114].

### 4.3. Antimicrobial Peptides (AMP) and Denture-Related Candida Stomatitis

In addition to cytokines and complement, epithelial cells have also been shown to produce a variety of antimicrobial peptides (AMP) in response to the presence of *Candida*, including β-defensins and cathelicidin (LL-37). Hundreds of AMPS are synthesized by epithelial cells and lymphocytes [115].

Among the AMPs, only LL37 (the sole member of the human cathelicidin family) showed a significant increase between a healthy state and DRCS, playing a role in the modulation of immune and inflammatory pathways [23]. However, AMP-LL37 was present in partially dentate patients suffering from inflammation (66%), while the healthy group was predominantly edentulous (95%). One explanation could be that the concentration of salivary AMP decreases with the loss of natural teeth. LL-37 also modulates the production of chemokines to promote chemotaxis. Moreover, LL-37 can induce transcription of CXCL8 alone and synergize with TNF-α-mediated expression of this chemokine [116].

### 4.4. Inflammosome

The inflammosomes participate locally in the host’s innate immune defense against *Candida*. They come into play through pathogen-associated molecular models (PAMPs) and damage-associated molecular models (DAMPs) [117]. The inflammasome NLRP3 (the nucleotide-binding domain and leucine-rich repeat receptors) is dependent on IL-1β responses, but also on molecules derived from pathogens such as *C. albicans* [118,119]. NLRP3 forms an assembly of characteristic proteins within macrophages [93,120]. The fungus is able to trigger pyroptosis and cell death within a macrophage that has engulfed it, by activating the NLRP3 inflammasome [92] (Figure 4). In the presence of *Candida*, activation by the NLRP3/ASC (apoptosis-associated speck-like protein) inflammasome causes an exaggerated innate reaction. This has the consequence of causing acute inflammation of the mucous membrane and promoting the dissipation of candidiasis. Locally, inflammasomes are essential for the antifungal defense of the host in vivo, but not in the hematopoietic compartment [121].

## 5. Denture-Related Candida Stomatitis and Adaptive Immunity

The adaptive immune system has the advantage of inducing immunological memory. This immune system monitors commensal organisms and reacts to the presence of fungal dysbiosis [122]. B and T cells are essential in this system: B cells produce antibodies and T helper (Th) cells support the defense of the mucosal host.

The chronicity of the presence of *Candida* on the surface of the mucosa increases the migration of B lymphocytes and IgA [123]. Millet and colleagues hypothesize that the B lymphocytes residing in the tissues and their antibody responses stabilize the commensal fungal community residing in the oral cavity. The specific Th17 cells only intervene in the presence of an overflow of commensals and severe disease. Another possible side effect in vivo of fungal colonization and an increase in B cells as well as antibody secretions is the shaping of the microbial community [124]. In the case of DRCS, an immune response involving T lymphocytes and monocytes has been demonstrated in the presence of *C. albicans* fungal antigens [125]. In this condition, the persistence of *C. albicans* in the oral cavity promotes the escape of regulatory T cells. However, using a murine model, *C. albicans* remains under the control of tissue residual memory (TRM) through Th17 cells [126].

## 6. Differences in Microbiology, Proteomics, and Biomarkers between DRCS Classes I, II, and III

### 6.1. Candida Species and Denture-Related Candida Stomatitis

Older studies and recent research show the participation of *Candida* spp. during the pathogenicity of DRCS [1,6]; however, there are differences between the three clinical classes of Newton. From 82 patients with DRCS, a total of 113 *Candida* spp. isolates were obtained. Based on Newton’s classification, this research shows that candidal species mixing leads to increased susceptibility to Newton type III DRCS, while type I presents isolates in which *C. albicans* dominates [19]. Recently, other authors did not find any significant difference between *Candida* counts on the surface of the palatal mucosa between two groups of 82 patients aged 20–85 years with and without DRCS [127]. Another investigation involving DRCS type II and III groups showed that 40% of the *Candida* proteins were from *C. glabrata*. Thus, a positive synergy has been suggested between *C. glabrata* and *C. albicans* in the pathogenesis of DRCS [33,128]. Consequently, the hypothesis was put forward that proteins from *C. glabrata* can influence the expression of inflammatory factors from *C. albicans*. The simultaneous reduction in *C. albicans* and *C. glabrata* offers a therapeutic possibility to fight DRCS [129]. The confirmation of these findings comes from a statistical analysis of the microbiological data from different types of Newton classes, showing that the presence of yeasts on the prostheses is increased from type I to type III, with extensive inflammation seen in Newton type III [130].

More recently, a study of 36 denture patients with stomatitis demonstrated the involvement of non-albicans *Candida*. Thus, *C. parapsilosis* and/or *C. tropicalis* were found, particularly, both on the prosthetic base and on the surface of the palatal mucosa [131]. The reliability of these results is called into question, on the one hand, because of the small size of the samples studied, not considering the influence of other microorganisms of the microbiota and, on the other hand, the multifactorial etiology of in vivo DRCS-related factors.

### 6.2. Proteomic Analysis of Saliva Reflects the Clinical Aspect of DRCS

Salivary proteins originating from salivary glands may predispose patients to DRCS by enhancing the survival of *Candida* or the formation of biofilm. Mucosal/epithelial-originated proteins may be a result of tissue damage due to the DRCS condition. Immunoglobulins (Igs), the major group of proteins identified, suggest the role of B-cell-mediated immunity, especially in type II DRCS.

DRCS types II and III are found to be similar in terms of human proteins. However, type III DRCS is characterized by a higher level of proteins derived from serum, such as ceruloplasmin, hemoglobins, serotransferrin, and albumin.

By contrast, DRCS type II exhibits high levels of immunoglobulin fragments compared with DRCS type III. This suggests an obvious vascular participation in the presence of DRCS type III. In the absence of DRCS, therefore, the innate immunity proteins appear to be sufficient to protect patients. Beyond a certain threshold, the presence of a high level of immunogloblins reveals an acute inflammatory response clinically diagnosed by the presence of DRCS type II, while serum proteins signify a chronic response in DRCS type III. The detection and analysis of these different proteins can help in the diagnosis and therapy of DRCS [129].

### 6.3. Markers of Denture-Related Candida Stomatitis

Most of the fungi of the oral cavity remain opportunistic, despite a considerable pathogenic arsenal. In the healthy host, they are not potent enough to overcome the normal, non-specific, or specific defense mechanisms. Additionally, the search for specific biomarkers of DRCS has proved difficult. Two proteins, including cystatin C and cystatin SN, are upregulated in the presence of prosthetic stomatitis of types II and III, whereas carbonic anhydrase 6 (CAH6) decreases.

CAH6 promotes bacterial growth, and it is also a salivary marker not specific to DRCS [130]. Other studies found that only 13 peptide masses are downregulated in DRCS compared with non-DRCS controls [132,133]. The increased detection of the cystatin C gene (CYTC) in the presence of *Candida* in association with DRCS constitutes a marker of the inflammatory response. Immunoglobulins have been detected, particularly in the presence of DRCS type III, which translates to a B cell immune response mediated by inflammation [129].

## 7. Saliva and Humoral Immune System

Saliva participates in the humoral immune system; it contains IgAs and macromolecules that can limit microbial growth (antimicrobial peptides) [134]. Salivary IgAs come from plasma cells in the salivary glands [135,136]. IgAs are the proteins of the mucosal immune system, the most sensitive and reactive with respect to the load of commensal microorganisms [137]. The IgA–*Candida* interaction attenuates the innate response by neutralizing the adhesion of fungi to the epithelial surface. An in vivo study argued that the crosslinking of pathogens mediated by IgA neutralizes the growth of organisms by preventing their separation after division [138]. IgA binding with *C. albicans* decreases secretions of CXCL8/IL-8, IL-1A, and IL-1B mediators while CCL20 is unaffected. Thus, the interaction of *C. albicans* with IgA attenuates the epithelial response (pro-inflammatory). On the other hand, additionally, in response to immune initiation of IgA, B lymphocytes migrate to the site colonized by the antigen [139].

This secretory IgA prevents and neutralizes the penetration of microorganisms into the epithelium [140]. In the presence of dysbiosis of the oral microbiota, in association with persistent colonization of *C. albicans*, the host can develop an adaptive response notably through the production of IgA. As a result, B lymphocytes within the oral cavity fight against fungal dysbiosis by participating in the maintenance of the commensal balance of *C. albicans* [123].

Thus, *C. albicans* in the commensal state does not induce an acute inflammatory response [76]. The secretory IgA antibody released by B cells constitutes the first line of protection against the surface antigen in relation to the innate and adaptive immunity of the host [141,142]. IgA of mucosal origin also controls the composition, both quantitative and qualitative, of commensal bacteria [143]. Regarding commensal *C. albicans*, its persistence stimulates the accumulation of B lymphocytes and plasma cells on the sites of fungal colonization [144,145], while recognition of fungi elicits an acute inflammatory response [88,146]. Hence, the hypothesis that IgA is a link between innate and adaptive immunity is advanced by some authors [142].

## 8. Neutrophils and Denture-Related Candida Stomatitis

The denture constitutes an “artificial stimulus,” which causes involuntary and perpetual activation of neutrophils [40]. In the case of older patients, *Candida* exacerbates the decreased function of the neutrophils [97]. In other words, the defense mechanisms induced by neutrophils depend on the individual predispositions to DRCS [96,97] (Figure 5). The possible cytotoxic effect of *C. albicans* on the survival of neutrophils and on their number has already been suggested by several authors [147,148]. However, the proliferation of oral polymorphonuclear neutrophils (PMNs) can have both negative and positive impacts on the integrity of the oral mucosa. Neutrophils react to interleukin chemokines produced by the activated epithelial cells and macrophages, such as IL-1α, IL-β, IL-8, IL-17, IL-22, IL-36, G-CSF, and β-defensin, after which migration to the *Candida*-infected mucosa occurs [149]. On the other hand, by releasing their potent mediators into the extracellular environment, PMNs can cause an imbalance in the oral microbiota [150].

Blood granulocytes from older individuals with or without DRCS had a reduced expression of surface markers, CXCR1, cell adhesion molecules, and CD62L (L. selectin). The corresponding integrin CD11b (adhesion molecule), CD62L (cell activity marker), and surface markers indicative of priming are altered as the neutrophils extravasate, which might impair chemotaxis and diapedesis of such cells [40]. Thus, the altered surface density of CD11b and CD62L can be caused by an ex vivo stimulus of blood neutrophils by the pro-inflammatory cytokine TNF [151].

Immunosenescence changes the phenotype neutrophils from the bloodstream as well as those entering the tissues. These modifications coupled with dysbiosis of the oral microbiota will promote DRCS [97,152,153]. DRCS neutrophils are not fully primed by the process of extravasion, but are more affected by the local inflammatory environment of the DRCS. Local salivary neutrophils can be more easily initiated than blood neutrophils, while the latter can still be initiated by TNF. The phenotype of these neutrophils is not unlike that of neutrophils in synovial fluid in the presence of arthritis, which may be initiated by TNF [154]. The potential of salivary neutrophils is initiated by local cytokines, which represents an important defense mechanism of the host against *C. albicans*. Indeed, the simple extravasion of neutrophils from systemic blood is not sufficient to respond to the inflammation of DRCS.

### Host Comorbidity and DRCS

The immune system allows the host to protect itself against a candidal attack. As a result, each patient using a removable denture presents a genetic specificity and, particularly in older people, a state of comorbidity likely to influence the occurrence of DRCS [155]. Thus, several pathologies and their treatments can increase a patient’s exposure to *C. albicans* infection [156]. Frequently the atrophic aspect of denture stomatitis appears to be related to a general pathology such as diabetes and hypertension [157].

Many other general pathologies favor the increase of *C. albicans* in the oral cavity: HIV/AIDS [158], cancer treatments [159], dental caries [160], and oral lesions (ulcerations, nodules, or granulomas) [161]. In these patients, the presence of a removable denture creates new niches for microbial colonization. On the other hand, the deficient immune function will promote the proliferation of *Candida* (*C. albicans* in 70%–80 % of cases), leading to oropharyngeal candidiasis in immunocompromised people [162]. Thus, the general diseases of the patient can interfere with wearing the removable prosthesis and taking medication can disrupt salivary secretion [124,163]. Several investigations, both in vivo and in vitro, have advanced the hypothesis that inflammation in DRCS is closely linked to the risks of comorbidity in relation to vascular function [164].

## 9. Conclusions

The pathology of DRCS is closely linked to the relationship between innate and adaptive host immunity. Beyond a certain threshold, yeast control depends mainly on the innate response, and the adaptive response then tends to limit mucosal damage. At this stage, host genetics may influence the immunopathology of DRCS. However, improving the management of patients with DRCS requires a better in vivo understanding of the transition between innate and adaptive immunities with candidal species. Ideally, in the presence of a removable prosthesis, the wearer’s immune system must contain the *Candida* invasion, while tolerating the cohabitation of different microbial commensals. Under these conditions, the local oral ecology and several general pathologies influence the diversity and quantity of the denture plaque. The epithelial cells of the oral mucosa in contact with the prosthesis present pattern recognition receptors (PRRs) that enable them to detect pathogen-associated molecular models (PAMPs), in particular several elements of the candidal envelope and bacterial components. This results in an inflammatory reaction that can trigger an immune response through pro-inflammatory and antimicrobial signaling pathways. Several systems allow the oral microbiome to influence host defenses. Thus, between the bacteria colonizing the surface of the oral mucosa, there is a constant confrontation aimed at eliminating certain organisms [165]. This phenomenon makes it possible to be protected against candidal invasion. However, in the presence of DRCS, colonization and penetration of a fungal load inside the prosthetic resin occur easily, without opposition. This fungal reservoir thus constituted partly explains the chronicity and aggravation of DRCS. Particularly for patients at risk (immunocompromised, older individuals), prevention remains the only way to control denture stomatitis. Furthermore, the fight against fungal and bacterial colonization followed by maturation of the denture biofilm involves periodic and topographical maintenance of the biotic mucosa and abiotic denture surfaces.

## Figures and Tables

**Figure 1 microorganisms-10-01437-f001:**
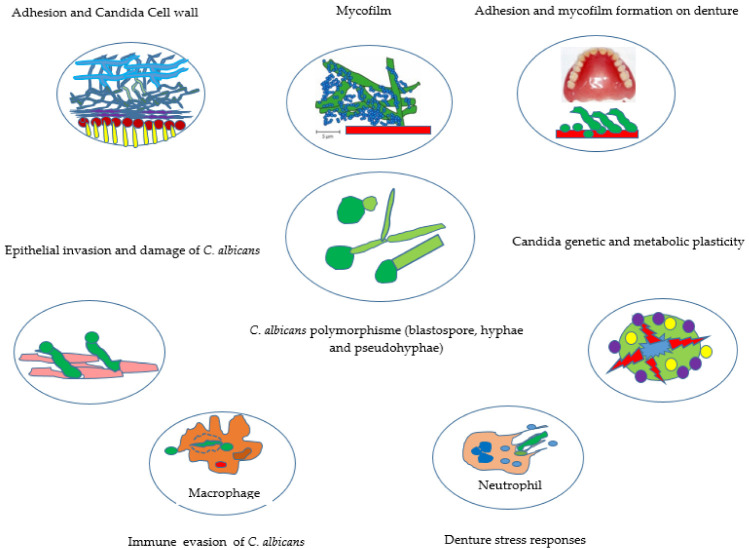
How does the presence of a removable prosthesis promote the virulence of *C. albicans*? Thanks to its morphological polymorphism, *C. albicans* can adapt to the prosthetic environment. After adhering to biotic and abiotic surfaces, these fungi can invade host cells and cause damage. It also has a genetic and metabolic potential enabling it to resist prosthetic stress as well as antifungal treatments. Finally, *Candida albicans* can evade the innate immune cells of the host. Indeed, *C. albicans* as a commensal has developed a resistance to the immune defenses of its host by evading on the one hand the mechanisms of recognition on its surface and on the other hand the process of phagocytosis of the macrophage.

**Figure 2 microorganisms-10-01437-f002:**
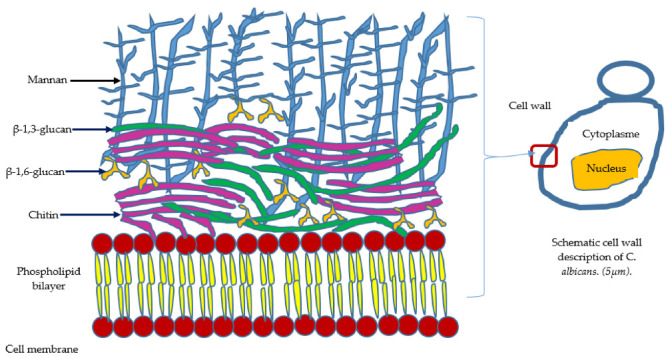
Pathogen-recognition receptors (PRRs) of the epithelial cell recognize different *C. albicans* PAMPs (pathogen-associated molecular patterns). Among PAMPS, on the outside of the *Candida* we find five layers within the cell wall. The outer layer is composed of C-linked proteins, mainly mannan (85%). Below is the β-1,6-glucan, the β-1,3-glucan, and the chitin layers. Underneath is a double layer of phospholipid framed by a membrane protein. PAMPs and mannan can be recognized by the PRRs of epithelial cells (Dectin-2, DC-SIGN, MINCLE, and TLR2/4/6). PAMPS, β-glucan, and chitin can be recognized by PRRs (Dectin-1, CR3, and NOD2, TLR4).

**Figure 3 microorganisms-10-01437-f003:**
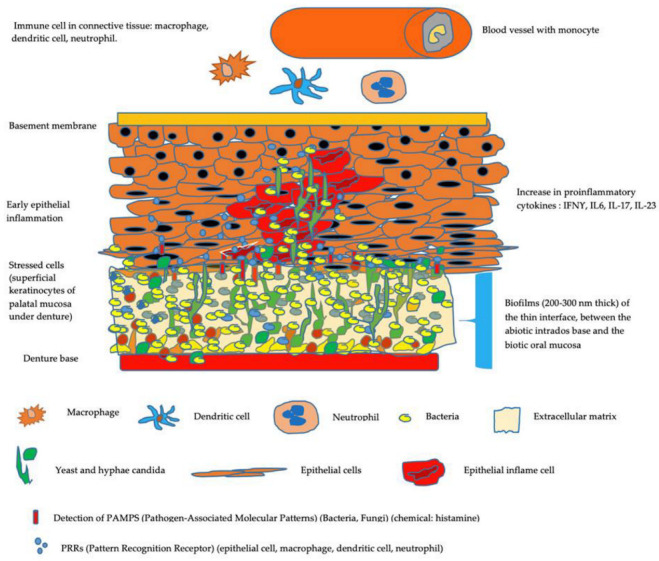
Model of palatal mucosa in the presence of early denture-related *Candida* stomatitis. The denture promotes a dysbiosis of the oral microbiome with an overgrowth of *C. albicans* and numerous commensal bacterial species. *C. albicans* in the form of hyphae crosses the epithelium of the mucosa and enables bacterial penetration. Bacterial species most frequently isolated with *Candida albicans* from these specific niches of the oral cavity are Streptococcus spp.: S. gordonii, S. mutans, S. salivaris. Saliva, moisture, nutrients, hyphal *Candida* morphotype, and the presence of commensal bacteria influence the architecture and virulence characteristics of mucosal fungal biofilms.

**Figure 4 microorganisms-10-01437-f004:**
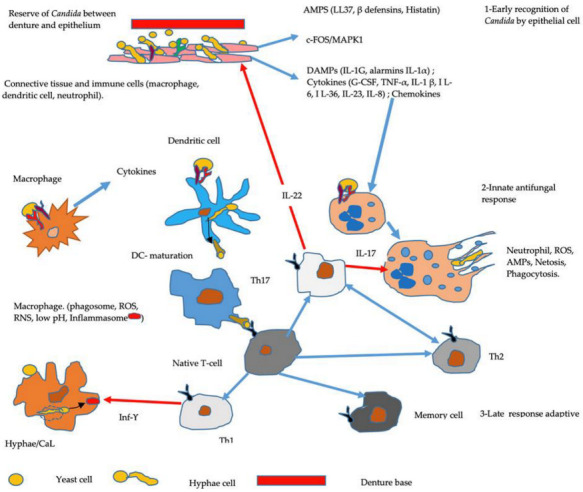
Schematic pathway of an early and late host immune response against the pathobiont *C. albicans* during mucosal invasion; early recognition of *Candida* (morphology of yeast and hyphal cells) by epithelial cells (neutrophils, macrophages, and dendritic cells) is efficient through pattern recognition receptors (PRRs), followed by innate and adaptive antifungal response [82].

**Figure 5 microorganisms-10-01437-f005:**
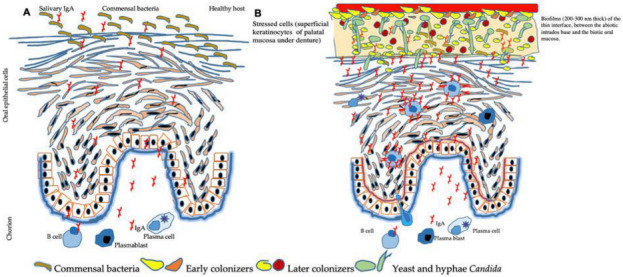
Under a removable prosthesis, the palatal mucosa can react by the appearance of signals, such as molecular models associated with damage (DAMP) and to damage cells or molecular models associated with pathogens (PAMPs). These surface signals alert different cells of the host deep down, which causes the activation of neutrophils. On the left (**A**), in the absence of *C. albicans*, salivary and tissue IgA regulate local immunity. These igAs participate in the homeostasis of the microbiota by binding to native bacteria. On the right (**B**), the presence of commensal *C. albicans* promotes the migration of mature B cells, plasmablasts, and plasma cells through the mucosa. IgA by binding to *C. albicans* decreases adhesion and slows down fungal colonization. Thus, the IgA will decrease the intensity of the pro-inflammatory response.

**Table 1 microorganisms-10-01437-t001:** The host can detect the invasive form from commensalism to infection by recognizing the passage from yeast to hyphae. *C. albicans* has the capacity to adapt its morphology according to the local conditions of the oral cavity (pH, θ, nutrients). Several forms—cellular, pseudohyphae, or true hyphae—allow *Candida* to proliferate and invade tissues. *C. albicans* can differentiate to form chlamydospores, enlarged thick-walled cells, under nutrient limitation, low temperature, and micro aerophilia.

Yeast-To-Hyphae Transition Chlamydospores	Polymorphism of *C. albicans* Fitness-Design-Plasticity	Dissemination to Invasion	Commensalism to Pathogens	References
Yeast (white, gray, opaque, 10 μm) 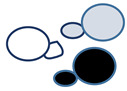	Unicellular round or ellipsoid	Colonization and dissemination	White (commensalism), gray (infectious), opaque (mating)	[47,48]
Hyphae (filaments), 10 μm–20 μm 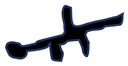	Multicellular	Invasion (the host can recognize the invasive form of *C. albicans*)	Penetrating form of *C. albicans* in epithelial cells	[12,49]
Pseudohyphae (filaments), 10 μm 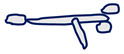	Multicellular separate septa	Infection	Infectious tissue (the host can use the yeast to hypha passage to distinguish between commensalism and infection)	[50]
Chlamydospores Suspensor cell 10 μm 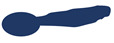	Thick-walled cells, suspensor cell with round cell at extremity	Not determined currently	Biological significance: survive under nutrient limitation, low θ	[44,45,46,51]

## Data Availability

Not applicable.

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
