# Peer review of "Host’s Immunity and Candida Species Associated with Denture Stomatitis: A Narrative Review"

_microorganisms, 2022, doi:10.3390/microorganisms10071437_

Round 1
Reviewer 1 Report
The paper "Host’s immunity and candida species associated with denture
stomatitis: a narrative review" is an interesting review about denture stomatitis due to Candida species and host's immunity. It would be appropriate to indicate in the introduction the possible correlation between Candida infection and potentially precancerous oral diseases ororal squamous cell carcinoma because C. albicans infection can contribute to the carcinogenesis process. (Hongal B.P., Kulkarni V.V., Joshi P.S., Karande P.P., Shroff A.S., Deshmukh R.S. Prevalence of fungal hyphae in potentially malignant lesions and conditions-does its occurrence play a role in epithelial dysplasia? J. Oral Maxillofac. Pathol. 2015;19:10–17. doi: 10.4103/0973-029X.157193, Astekar M., Roy S.K., Sapra G., Chitlangia R.K., Raj N. Evaluation of candidal species among individuals with oral potentially malignant disorders and oral squamous cell carcinoma. J. Oral Maxillofac. Pathol. 2019;23:302. doi: 10.4103/jomfp.JOMFP_111_18,
Author Response
Reviewer #1
Dear Reviewer,
Thank you for the excellent review of our manuscript “"Host immunity and Candida species associated with dental stomatitis: a narrative review”
Comments and Suggestions for Authors
The paper "Host’s immunity and candida species associated with denture stomatitis: a narrative review" is an interesting review about denture stomatitis due to Candida species and host's immunity. It would be appropriate to indicate in the introduction the possible correlation between Candida infection and potentially precancerous oral diseases or oral squamous cell carcinoma because C. albicans infection can contribute to the carcinogenesis process.
Our response: Thank you for this valuable comment, we apologize for this mistake. This information has been added in the introduction “In addition, C. albicans infection may contribute to the process of carcinogenesis, so a possible correlation between Candida infection and potentially precancerous oral diseases such as oral squamous cell carcinoma has been suggested by several authors” (page 2; line 62-65). We thank you for suggesting references that have been included in the bibliography (N° references: 14-16).
Reviewer 2 Report
The authors present a review on the Candida species and Host's immunity interactions in the context of denture stomatitis.
The work focuses on C. albicans, the most prominent representative of candidal infections, followed by C. glabrata, neglecting other candida species, such as C. tropicalis and C. parapsilosis. Adding a piece of brief information regarding the non-albicans prevalence on DRCS, as suggested by the title, and discussing differences with C. albicans DRSC would enrich the work.
The introduction should be improved to be more engaging. It also needs to be more concise and include recent literature, particularly regarding clinical-associated information (i.e., prevalence).
A figure with the schematic process described in section 3, "Denture plaque and C. albicans virulence," should be included.
The figures' quality and clarity must be improved.
Some clinical data examples should include and discuss more than one reference when available. In some cases, the authors refer to a single reference to generalize a fact (i.e., lines 70-72, lines 120-128).
Author Response
Dear Reviewer,
Thank you very much for the interest given to the present article and the excellent review of our manuscript. This has helped us substantially to improve the quality of this manuscript. We totally agree to say that English needs revision. We have now completely revised the manuscript. We took in consideration all your suggestions in the attachment. We made a point-by-point response, and we highlighted the changes to our manuscript. We hope that the revised version will meet your approval.
Thank you,
Sincerely,
Comments and Suggestions for Authors
The authors present a review on the Candida species and Host's immunity interactions in the context of denture stomatitis. The work focuses on C. albicans, the most prominent representative of candidal infections, followed by C. glabrata, neglecting other candida species, such as C. tropicalis and C. parapsilosis. Adding a piece of brief information regarding the non-albicans prevalence on DRCS, as suggested by the title, and discussing differences with C. albicans DRSC would enrich the work.
Our response: Thank you for this valuable suggestion, we totally agree for this comment, and we have added the information regarding the non-albicans prevalence on DRCS to the text. This sentence has been added “More recently, a study of 36 denture patients with stomatitis has demonstrated the involvement of non-albicans Candida. Thus, C parapsilosis and/or C tropicalis were found, particularly, both on the prosthetic base and on the surface of the palatal mucosa” (page 13; lines 485-487).
This required the addition of a new reference “Lurdete M. R. ; Simone S. -P. ; Silveira-Gomes F. ; Renata A. E. ; Marques-da-Silva S. H. Isolation of Candida spp. from denture-related stomatitis in Pará, Brazil. Braz J Microbiol, 2018, 148-151”.
The introduction should be improved to be more engaging. It also needs to be more concise and include recent literature, particularly regarding clinical-associated information (i.e., prevalence).
Our response: We thank you for this comment, whose answers have enabled us to improve the quality of the manuscript. Thus, we have added the following sentence on the prevalence ""The prevalence of this pathology is preponderant among hospitalized elderly people [ ] smokers[ ] and people with associated affections such as diabetes[ ]" (page 1; 41-43). This necessitated the addition of three new references on the subject:
Razia Z. A.; Kimmie-Dhansay F.Prevalence of Denture-Related Stomatitis in Edentulous Patients at a Tertiary Dental Teaching Hospital. Front Oral Heath. 2021,1,2,772679.
Farimah S.; Parvin K.; Hamid H.; Saadat M.; Pouya A. The prevalence of denture stomatitis in cigarette and hookah smokers and opium addicts: findings from Rafsanjan Cohort Study. BMC Oral Health. 2021, 17, 21,455.
Martorano-Fernandes L.; Dornelas-Figueira L. M.; Marcello-Machado R. M.; de Brito Silva R.; Marcela B. M.; Lucianne C. M,; Altair A. D. B. Cury. Oral candidiasis and denture stomatitis in diabetic patients: Systematic review and meta-analysis. Braz Oral Res. 2020,21, e113.
Concerning the necessity for the introduction to be more concise we delete the phrase below
“In addition, many general conditions promote the development of Candida; dietary defi-ciency, a deficit in salivary secretion, tobacco, and alterations in the host's immune de-fenses, either of hereditary or acquired origin, are most often blamed [10]”.
A figure with the schematic process described in section 3, "Denture plaque and C. albicans virulence," should be included.
Our response: Thank you for your comment. Indeed, we have added a new figure as mentioned, in your comment on page 5, lines178-214) . It is the figure mentioned below (see in the manuscript, page 5, lines178-214).
Figure 1. How does the presence of a removable prosthesis promote the virulence of C. albicans? Thanks to its morphological polymorphism, C. albicans can adapt to the prosthetic environment. After adhering to biotic and abiotic surfaces, these fungi can invade host cells and cause damage. It also has a genetic and metabolic potential enabling it to resist prosthetic stress as well as antifungal treatments. Finally, C. albicans can evade the innate immune cells of the host. Indeed, C. albicans as a commensal has developed a resistance to the immune defenses of its host by evading on the one hand the mechanisms of recognition on its surface and on the other hand the process of phagocytosis of the macrophage.
The figures' quality and clarity must be improved.
Our response: We agree with you on this point. Indeed, the figures have been made from Word, so the original drawings have been used. This has undoubtedly had an effect on the resolution. However, we have improved their definition to meet the requirements of the review. We hope that these new figures meet your requirements. In addition, an update has been made to the figures, resulting in a new numbering with the addition of a new figure.
Some clinical data examples should include and discuss more than one reference when available. In some cases, the authors refer to a single reference to generalize a fact (i.e., lines 70-72, lines 120-128).
Our response: Our response: Thank you for your comment. We have added a new reference “Nikawa H. ; Hamada T. ; Yamamoto T. Denture plaque--past and recent concern. J Dent. 1998, 26,299-304” to address this comment from lines 70-72. The same applies to lines 120-128 where we have added this paragraph (page 3; lines 137-144) "On the contrary, another study, based on the culture of Candida spp, shows in a population of 123 patients fitted with orthopaedic devices a high candidal population that can lead to an imbalance of the oral microbiome [ ] . Confirmation is provided by Fujinami et al in a sample of 18 dentures wearers (mean age, 80.3 years). Based on measurements of C. albicans DNA concentrations and bacteria by real time PCR this study shows the abun-dance of the genera Streptococcus, Lactobacillus, Rothia and Corynebacterium on the surface of removable dentures compared to dental plaque. C. albicans was positively correlated with these acidogenic bacteria".
This required the addition of two new references.
O’Donnell, L. E.; Robertson, D.; Nile, C. J.; Cross, L. J.; Riggio, M.; Sherriff, A.; Bradshaw, D.; Lambert, M.; Malcolm, J.; Buijs, M. J.; et al. The oral microbiome of denture wearers is influenced by levels of natural dentition. PLoS ONE. 2015, 10, e0137717.
Fujinami W.; Nishikawa K.; Ozawa S.; Hasegawa Y.; Takebe J. Correlation between the relative abundance of oral bacteria and Candida albicans in denture and dental plaques. J Oral Biosci. 2021, 63,175-183.
Reviewer 3 Report
Dear Authors
Denture-related Candida stomatitis is characterized by a lesion which is almost invariably asymptomatic and usually affects the hard palate's mucosa underlying a removable, partial or total, dental prosthesis. It has a multifactorial aetiology and a high mean prevalence in population-based series of randomly selected patients with removable prostheses.
In recent years, the use of immunosuppressive chemotherapy and the emergence of HIV-infection have increased the interest in infections caused by Candida spp. Additionally the presence of a removable prosthesis within the oral cavity creates a favourable environment for proliferation of microorganisms mainly between the supporting mucosa and the fitting surface of the denture, allowing the change of a commensal microorganism into a pathogenic one.
Thus, this is a very current and wide-ranging topic with clinical implications, that needs to be studied so that it can be better understood and easier to manage.
The authors did an extensive review in which the host immune response to Candida infection in the oral cavity, in its different aspects, were evidenced allowing to optimize the diagnosis and the therapy of DRCS.
The article is well written and is easy to read. I just found the two following mistakes:
Line 193: Here it should be point 3.4 and not 3.3, as this (3.3) is already written above.
Line 395: I think you want to write “has” instead of “hass”.
The high number of references is justified because this is a review article. The references are related to recent, current and relevant works.
Best regards
Author Response
Dear Reviewer,
We thank you for your interest in our manuscript entitled "Host immunity and Candida species associated with dental stomatitis: a narrative review". Your remarks and comments have allowed us to make the necessary corrections. We have corrected your two remarks point by point. These modifications appear in the new version of the article.
Comments and Suggestions for Authors
Denture-related Candida stomatitis is characterized by a lesion which is almost invariably asymptomatic and usually affects the hard palate's mucosa underlying a removable, partial or total, dental prosthesis. It has a multifactorial aetiology and a high mean prevalence in population-based series of randomly selected patients with removable prostheses.
In recent years, the use of immunosuppressive chemotherapy and the emergence of HIV-infection have increased the interest in infections caused by Candida spp. Additionally the presence of a removable prosthesis within the oral cavity creates a favourable environment for proliferation of microorganisms mainly between the supporting mucosa and the fitting surface of the denture, allowing the change of a commensal microorganism into a pathogenic one.
Thus, this is a very current and wide-ranging topic with clinical implications, that needs to be studied so that it can be better understood and easier to manage.
The authors did an extensive review in which the host immune response to Candida infection in the oral cavity, in its different aspects, were evidenced allowing to optimize the diagnosis and the therapy of DRCS.
The article is well written and is easy to read. I just found the two following mistakes:
Line 193: Here it should be point 3.4 and not 3.3, as this (3.3) is already written above.
Line 395: I think you want to write “has” instead of “hass”.
The high number of references is justified because this is a review article. The references are related to recent, current and relevant works.
Concerning the two errors we have made the following changes:
Our response Error 1: We have corrected the one in page 5, line no. 193 (old version of the manuscript) by replacing the notation 3.3. by 3.4. in the new manuscript version page 6, line 247. Thus, the notation of the paragraph: 3.3. The immune evasion of C. albicans becomes 3.4. Immune evasion of C. albicans.
Our response Error 2: The same applies to page 11, line 395 (old version of the manuscript). We have replaced "hass" with "has". Thus, the sentence line 395 (old version of the manuscript) becomes « The adaptive immune system has the advantage of inducing immunological memory» in the new version on page 12, line 449.